# Modeling Hydro-Dynamics in a Harbor Area in the Daishan Island, China

**Yuting Li** [1,2,3], **Zhiyao Song** [1,2,3,*], **Guoqiang Peng** [1,2,3] , **Xuwen Fang** [4], **Ruijie Li** [5,6,*], **Peng Chen** [1,2,3] **and Haoyuan Hong** [1,2,3]

[1] Key Laboratory of Virtual Geographic Environment, Nanjing Normal University, Nanjing 210023, China; yutingli123@163.com (Y.L.); geop1987@gmail.com (G.P.); chenpeng09cp@163.com (P.C.); hong_haoyuan@outlook.com (H.H.)
[2] State Key Laboratory Cultivation Base of Geographical Environment Evolution (Jiangsu Province), Nanjing 210023, China
[3] Jiangsu Center for Collaborative Innovation in Geographic Information Resource Development and Application, Nanjing 210023, China
[4] Anhui Transport Survey & Design Institute Co., Ltd., Hefei 230000, China; fangxuwen321@163.com
[5] Key Laboratory of Coastal Disaster and Defence, Ministry of Education, Hohai University, Nanjing 210098, China
[6] College of Oceanography, Hohai University, Nanjing 210098, China
* Correspondence: zhiyaosong@163.com (Z.S.); rjli@hhu.edu.cn (R.L.)

**Abstract:** This study presents an incorporation and application of a two-dimensional, unstructured-grid hydrodynamic model with a suspended sediment transport module in Daishan, China. The model is verified with field measurement data from 2017: water level, flow velocities and suspended sediment concentration (SSC). In the application on the Daishan, the performance of the hydrodynamic model has been satisfactorily validated against observed variations of available measurement stations. Coupled with the hydrodynamic model, a sediment transport model has been developed and tested. The simulations agreed quantitatively with the observations. The validated model was applied to the construction of breakwaters and docks under a different plan. The model can calculate the flow field and siltation situation under different breakwater settings. After we have analyzed the impact of existing breakwater layout schemes and sediment transport, a reasonable plan will be selected. The results show that the sea area near the north of Yanwo Shan and Dongken Shan has a large flow velocity exceeding 2.0 m/s and the flow velocity within the isobath of 5 m is small, within 0.6 m/s. According to the sediment calculation, the dock project is feasible. However, the designed width of the fairway should be increased to ensure the navigation safety of the ship according to variation characteristics of cross flow velocity in channel.

**Keywords:** suspended sediment concentration; numerical simulation; breakwater; the dock project

## 1. Introduction

Suspended sediment concentration (SSC) in tidal estuaries can affect the physical, chemical and biological environment of the water column [1–3]. Suspended sediments can change light attenuation and affect the cycle of nutrients, organic micropollutants and heavy metals [4,5]. Therefore, knowledge of suspended sediment dynamics is essential for quantifying the fluxes of ecologically and chemically important substances and determining the fate of pollutants [6,7].

Marine coastal areas are considered the most productive and dynamic ecological resources and pave the way for massive socioeconomic activities in the world [8–10]. The construction of coastal engineering could have a great influence on the hydrodynamics, seabed evolution and ecological

environment near the project. Understanding and predicting the influences of coastal engineering on sediment transport can provide a basis for understanding the changes of hydrodynamics.

Due to the difficulty of measuring the sediment dynamics in the laboratory and in situ, one way to study the sediment dynamics is with a numerical model [11–14]. Numerical modeling has become more attractive because it overcomes the problems of scaling effects from which physical modeling often suffers. The higher costs and longer time involved in constructing and running a physical model can be avoided [15]. Numerical models have many advantages [16]. Lots of research has been undertaken, ranging from 1D to 3D. 1D models [17] have been considered because of their low computational cost and data requirements. However, under the presence of complex topography, the use of 2D/3D hydrodynamic models is required. 3D models are used in the following studies: Van Rijn [18], Bijvelds et al. [19], Wang and Jia [20], Wu et al. [21], Fang and Rodi [22], Chen et al. [11], Lu et al. [23]. The negative qualities of 3D models are: (i) the high number of equations they require and (ii) the high computational costs [24].

Conversely, the 2D models are computationally efficient since the number of equations is limited. In the previous work, many two-dimensional sediment mathematical models were established (e.g., Wang and Adeff [25], Li et al. [26], Ke et al. [27] and Petti et al. [28]). Thus, in this work a 2D mathematical model is considered. It is important to have an improved understanding of estuarine hydrodynamics and suspended sediment transport [29]. Our goal was to build on existing studies through a case study of the partially mixed sea area near Daishan, China, in which the role of tide-induced sediment processes and model parameterizations of those processes can be further illustrated. However, in a natural river it is at present not easy to express the aforementioned factors with a general and fully accurate formula [30]. This is because each marine coastal area, even each river segment, has individual characteristics that depend on the complex boundary conditions and incoming flow and sediment from upstream.

In this paper, a numerical simulation is used to analyze the flow field and sedimentation after the implementation of the dock project. These numerical studies are of great help to the understanding of suspended particulate materials (SPM) and sediment transport and distribution in the sea near Daishan Island. The influence of the existing layout scheme of the breakwater on the water flow and sediment transport is analyzed and a reasonable solution is selected in order to try to reduce the influence of human activities.

The paper is organized as follows. The study area and data used are described in Section 2. In Section 3, the model development and governing equations are reported. The model verification is presented in Section 4. Details about the engineering application, calculated with a small-scale mathematical model, are given in Section 5. All the results obtained in this paper are discussed in Section 6. Finally, in Section 7, the conclusions of the study are drawn.

## 2. Study Area and Data Used

The study area, Daishan Island, is situated in Zhejiang province, China between 30°13′ N and 30°21′ N and 121°3′ E and 122°13′ E. It is connected to the Yangtze River estuary in the north, Hangzhou Bay in the west, Donghai in the east and Zhoushan Island in the south. It is located within the international route outside the Yangtze River Estuary (Figure 1). The study area is the second-largest island in Zhoushan, which overall covers 119.3 km$^2$. Daishan Island belongs to the monsoon oceanic climate in the southern margin of the northern subtropical zone. There are many types of landforms such as beaches, tidal flats and low mountains on the island. They are often affected by alternating cold and warm air and frequent severe weather.

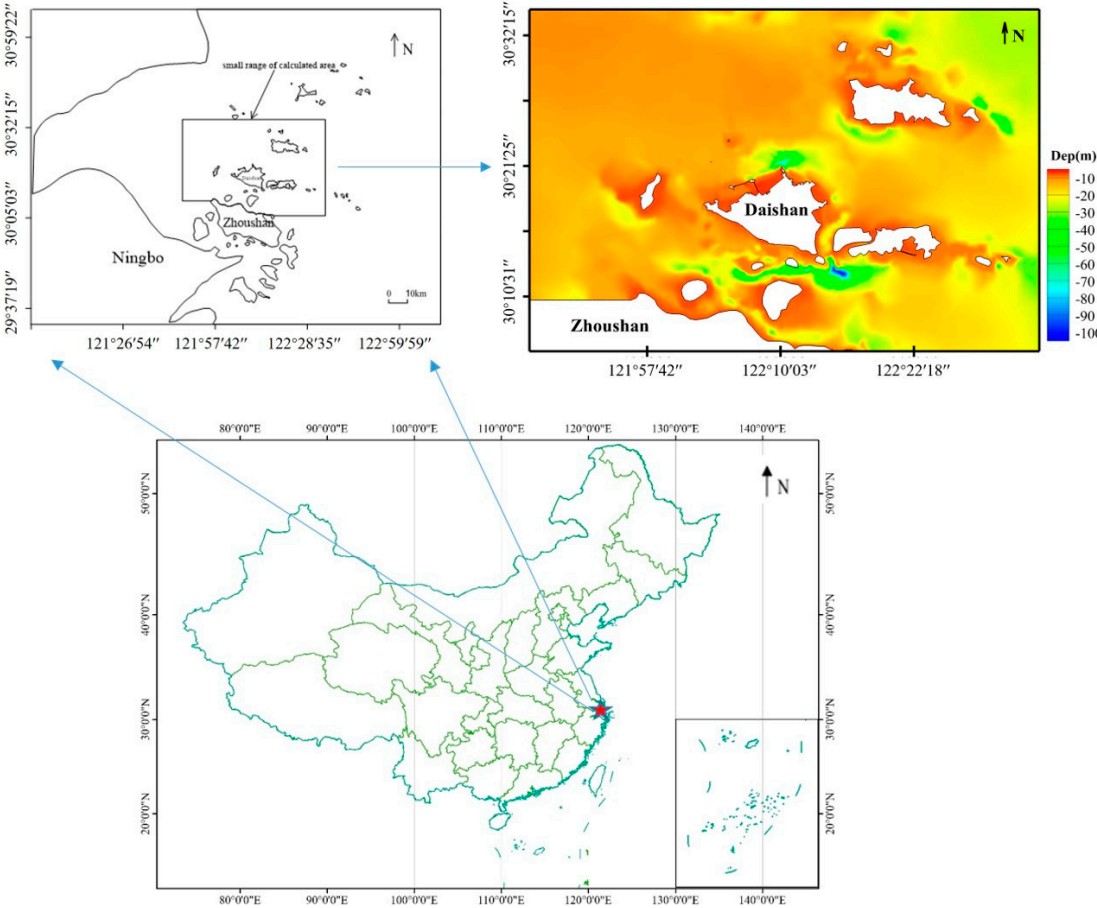

**Figure 1.** Topographic map of simulated domain.

## 2.1. Weather Conditions

The average annual precipitation of Daishan is 1189.8 mm, the most abundant year is 1636.1 mm (2002) and the driest year is 701.3 mm (1967). The precipitation is bimodal in the year and the peak is in June. The monthly rainfall is usually 14.4% of the whole year, mainly formed by plum rain. In September, the rainfall in this month generally accounts for 13.3% of the whole year. Mainly due to typhoon and rain.

## 2.2. Tidal Current Conditions

The tide in engineering sea area is dominated by the tidal of M2 constituent. The engineering sea area has an irregular semidiurnal tide and the tidal asymmetry is obvious. A shallow tide exists in all three temporary tidal stations. The tidal current movement in the engineering sea area is reciprocating flow. The measured maximum tidal current velocity is 2.28 m/s and the flow direction is 216°; the maximum ebb current velocity is 2.52 m/s and the flow direction is 77°; the vertical average maximum tidal current velocity is 2.17 m/s and the flow direction is 237°; the vertical average maximum ebb current velocity is 2.34 m/s and the flow direction is 78°.

## 2.3. SPM Conditions

The sediments in the engineering sea area are mainly fine-grained silt and clay, of which about 69% are silt, 25% clay and a small amount sand (about 6%). The median particle size range is 3.72–6.51 mm, with an average of 5.17 mm. The spatial distribution of the median particle size of the sediments is eastern—rough and western—fine. The maximum sediment concentration is 1.874 kg/m$^3$ and the

minimum sediment concentration is 0.171 kg/m$^3$. The maximum vertical averaged suspend sediment concentration is 1.495 kg/m$^3$ and the minimum value is 0.342 kg/m$^3$.

The tide level and tidal current calculated by the model are verified by the tide level data of the measured stations W1, W2 and W3. The flow velocity and SSC data are measured by stations SW1–SW6. The station distribution is shown in Figure 2 and station coordinates are shown in Table 1.

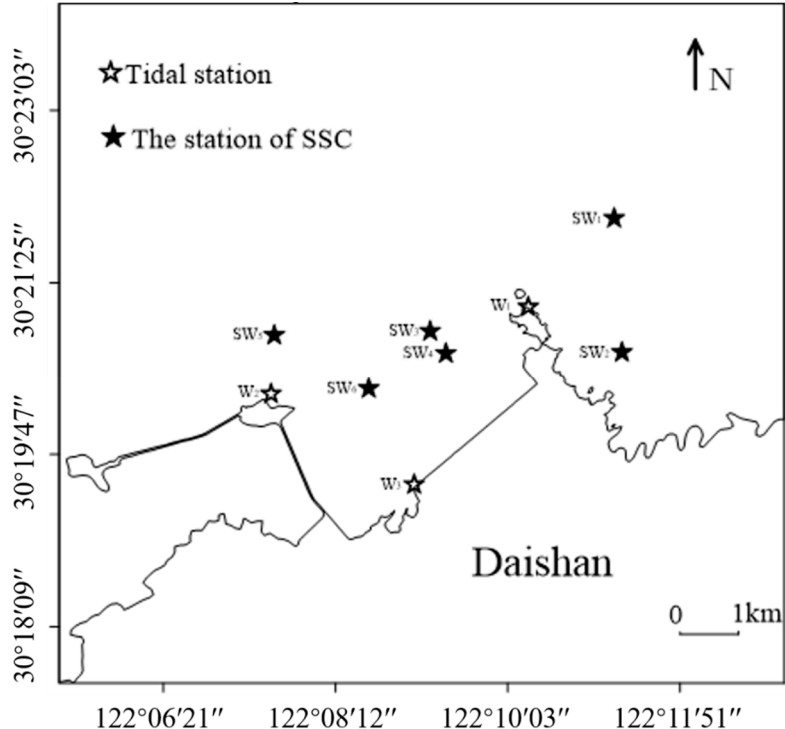

**Figure 2.** The engineering station of northern Daishan Island.

**Table 1.** Coordinates of hydrological stations in the engineering sea area.

| Station | Actual Observation Site | | Observation |
| --- | --- | --- | --- |
| | WGS-84 Coordinates | | |
| | N | E | |
| SW1 | 30°22′00.6″ | 122°11′13.5″ | SSC, Bottom material |
| SW2 | 30°20′44.9″ | 122°11′19.2″ | SSC, Bottom material |
| SW3 | 30°20′56.1″ | 122°09′14.1″ | SSC, Bottom material |
| SW4 | 30°20′43.6″ | 122°09′24.4″ | SSC, Bottom material |
| SW5 | 30°20′53.2″ | 122°07′32.4″ | SSC, Bottom material |
| SW6 | 30°20′23.0″ | 122°08′34.3″ | SSC, Bottom material |
| W1 | 30°21′10.2″ | 122°10′17.9″ | Tide level |
| W2 | 30°20′19.5″ | 122°07′30.6″ | Tide level |
| W3 | 30°19′29.3″ | 122°09′08.8″ | Tide level |

## 3. Model Development and Equations

### 3.1. The Hydrodynamic Model

This section provides a brief description of the proposed method. The model adopts the vertical average two-dimensional shallow water equation. ADI (Alternating Direction Implicit) is used to discretize the mass and momentum equations [31]. A DS (Double Sweep) scheme is adopted to discretize the resulting matrix. Then the central difference scheme is used for each differential term and coefficient. It is worth noting that, in order to prevent the mass and momentum distortion that may

occur in the discrete process, we need to control the truncation error of the Taylor series, expanded to achieve to 2nd or 3rd order accuracy.

A cell-centered finite volume method (FVM) is used to solve the numerical model with time-varying elevation boundary conditions. The space domain is discretized using an unstructured mesh with triangular elements. The detail of the cell-centered FVM are provided by Anastasiou and Chan, Jiwen and Ruxun and Benkhaldoun and Seaïd (2010) [32–34].

The 2D mathematical model estimates the impact of the new layout on the local SSC and tidal current. The 2D mathematical model adopted in the case study is a MIKE21 hydrodynamic (HD) model (DHI, 2007) [35] that solves the 2D nonlinear shallow water equations (SWE), resulting in the temporal evolution of the flow field over the spatial domain. The 2D SWE can be described by a continuity equation as follows:

$$\frac{\partial d}{\partial t} + \frac{\partial}{\partial x}(ud) + \frac{\partial}{\partial y}(vd) = 0 \tag{1}$$

In Equation (1), $d$ is the total water depth; $d = h + \eta$; $\eta$ is the water level; $h$ is the hydrostatic depth; and $u$ and $v$ are current velocities at the $x$, $y$ direction, respectively.

Momentum equations are as follows:

$$\frac{\partial u}{\partial t} + u\frac{\partial u}{\partial x} + v\frac{\partial u}{\partial y} - fv = -g\frac{\partial \eta}{\partial x} + \frac{\partial}{\partial x}\left(\varepsilon_{xx}\frac{\partial u}{\partial x}\right) + \frac{\partial}{\partial y}\left(\varepsilon_{xy}\frac{\partial u}{\partial y}\right) - \frac{gu}{C_z^2 d}\left(u^2 + v^2\right)^{1/2} \tag{2}$$

$$\frac{\partial u}{\partial t} + u\frac{\partial v}{\partial y} + v\frac{\partial v}{\partial x} + fu = -g\frac{\partial \eta}{\partial y} + \frac{\partial}{\partial x}\left(\varepsilon_{yx}\frac{\partial v}{\partial x}\right) + \frac{\partial}{\partial y}\left(\varepsilon_{yy}\frac{\partial v}{\partial y}\right) - \frac{gv}{C_z^2 d}\left(u^2 + v^2\right)^{1/2} \tag{3}$$

In Equations (2) and (3), $g$ is the acceleration of gravity; $f$ is the Coriolis parameter; $C_z$ is the Chezy coefficient; and $\varepsilon_{xx}$, $\varepsilon_{xy}$, $\varepsilon_{yx}$ and $\varepsilon_{yy}$, are the coefficients of eddy viscosity obtained by the Smagorinsky [36] model in different directions.

### 3.2. The Suspended Sediment Transport Model

The 2D mathematical model adopted in the case study is a MIKE21 mud transport (MT) model (DHI, 2007). The governing equation for the total suspended sediment concentration is as follows:

$$\frac{\partial dC}{\partial t} + \frac{\partial duC}{\partial x} + \frac{\partial dvC}{\partial y} = \frac{\partial}{\partial x}\left(\varepsilon xd\frac{\partial C}{\partial x}\right) + \frac{\partial}{\partial y}\left(\varepsilon yd\frac{\partial C}{\partial y}\right) + F_c \tag{4}$$

where $C$ denotes SSC; $u$ and $v$ are the velocity components in the $x$, $y$ directions; $\varepsilon_x$ and $\varepsilon_y$ are diffusion coefficient; and $F_c$ is the suspended sediment flux, given by the following:

$$F_c = \begin{cases} \alpha \omega C(\tau_b/\tau_d - 1) & \tau_b \leq \tau_d \\ 0 & \tau_d < \tau_b < \tau_e \\ M(\tau_b/\tau_e - 1) & \tau_b \geq \tau_e \end{cases} \tag{5}$$

In Equation (5), $\alpha$ is the settlement probability of sediment, varying from 0.67 to 0.84; $\omega$ is the settling velocity of SPM; $M$ is the coefficient of scouring, which is determined by sediment model verification; $\tau_b$ is bed shear stress in $\tau_b = \rho U_*^2$ ($\rho$ is the water density and $U_*$ is the shear velocity); $\tau_e$ is the critical shear stress for sediment resuspension in $\tau e = \rho U_{*e}^2$ ($U_{*e}$ is the critical shear velocity for sediment resuspension); $\tau_d$ is the critical shear stress for sediment deposition in $\tau_d = \rho U_{*d}^2$; $U_{*d}$ is the critical shear velocity for sediment deposition. The details of $\alpha$, $M$, $U_{*d}$ and $U_{*d}$ are provided by Bosa et al. [37], Hu et al. [38] and Xie et al. [39].

Topographical change equation:

$$\gamma_0\frac{\partial Z_b}{\partial t} = -F_c \tag{6}$$

In Equation (6), $\gamma_0$ is the dry density of sediment; $Z_b$ is the bed elevation.

With no consideration of offshore wind action, the land boundary can be expressed as:

$$Vn(x, y, t) = 0 \qquad (7)$$

The water level at the open boundary is calculated by the tidal current model of Donghai, China.

The normal zero flux boundary condition is adopted for the suspended sediment boundary condition. The boundary conditions for suspension of sand are as follows:

$$\begin{cases} C(x, y, t)\,|_\Gamma = C^*(x, y, t) \\ \frac{\partial C}{\partial t} + \frac{\partial (Cu)}{\partial x} + \frac{\partial (Cv)}{\partial y} = 0 \end{cases} \qquad (8)$$

In Equation (8), $\Gamma$ is the open boundary of waters; $C^*(x, y, t)$ is the suspended sediment concentration, which is known in this paper.

## 4. Model Setup

The calculation area of the model is shown in Figure 3. In the north, the East China Sea to the south of the Yangtze Estuary and the east of Nanhui Zui are the boundaries of the computational domain. In the south, the East China Sea to the south of Liuheng Island is the open boundary of the calculation area. About 40 km east of Shengshan island is the eastern open boundary of the calculated area. On the west side, Hangzhou Bay is considered the open boundary of the computational domain and the other boundaries are solid boundaries.

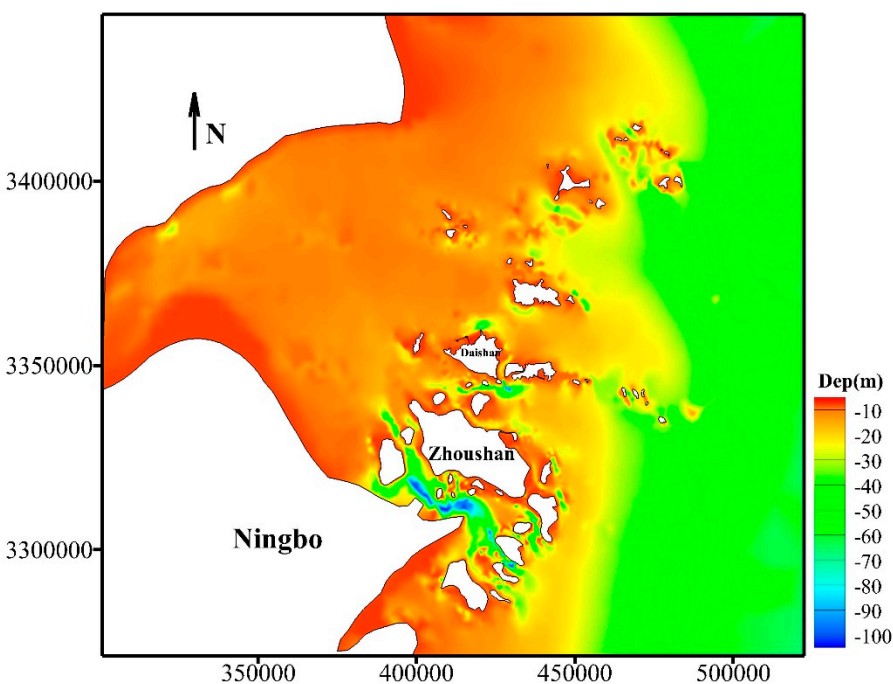

**Figure 3.** The study area of relief map.

In the article by Pan et al. [40], hydrodynamics provides descriptions of the hydrodynamic model mesh setup, parameter calibrations and verifications and a satisfactory performance in predicting surface water elevations and currents to ensure accuracy in the sediment transport modeling in this study.

The unstructured, triangular grid system was developed to cover the Daishan Island study area (Figure 4). The domain was discretized by a triangular, unstructured grid of 28,987 cells and 15,468 nodes. The minimum mesh size is 50 m. Using the 1800-s time step, model simulations were conducted to simulate the tidal current from 14 March to 14 April 2017.

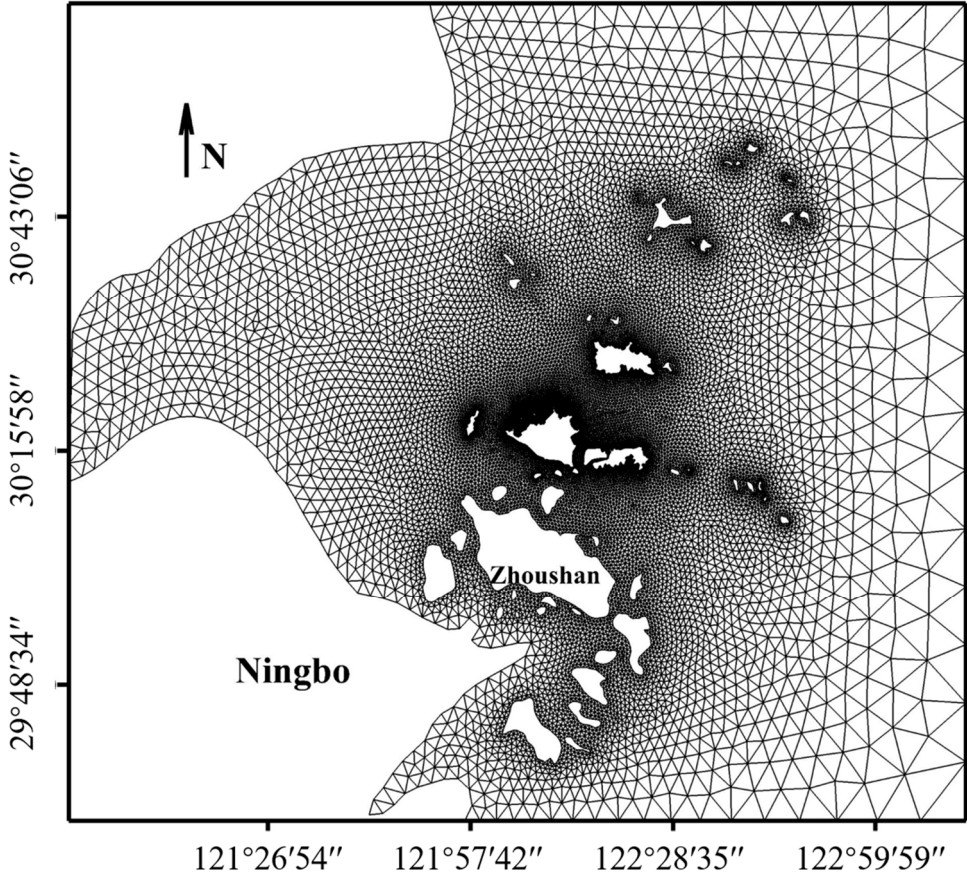

**Figure 4.** An unstructured, triangular mesh for the study area.

Several parameters have been calibrated to minimize the deviations between simulated and measured data within acceptable ranges of accuracy. The calibrated model parameters included a Manning's *n* of 0.015 $m^{1/3}$/s–0.020 $m^{1/3}$/s for ocean areas. In this paper, the Manning's *n* of the study area is taken as 0.02 $m^{1/3}$/s. The horizontal vortex viscosity coefficient is calculated by the Smagororinsky (1963) formula considering the sub-scale grid effect, which is taken as 0.28; and a Courant-Friedric-Levy (CFL) number of 0.7. The CFL condition is used to ensure the model stability in the MIKE 21 HD model. This condition states that the CFL number must be less than 1 in order for the model to be stable. Kregting and Elsäßer [41] used a MIKE21 hydrodynamic model with a critical CFL of 0.7 for the prediction of engineering applications within Strangford Lough.

## 5. Validation of the Model

Model verification is an important procedure for ascertaining the model's accuracy before the numerical experiment stage [42]. An accurate dataset is required to calibrate the numerical model and validate its capacity for predicting water level, tidal currents and SSC. In this study, a dataset consisting of water level, flow velocity and SSC, measured from 00:00 on 21 March 2017 to 00:00 on 7 April 2017, was used for model verification. The process of tidal level change of large, medium and small tides has been used for validation. Not all are listed due to space limitations.

### 5.1. Validation of Water Level

Model predictions of water levels were compared with observations taken in the field. Figure 5 shows model and measured water level data of observations at (a) W1, (b) W2 and (c) W3. Due to space limitations, they are not listed one by one.

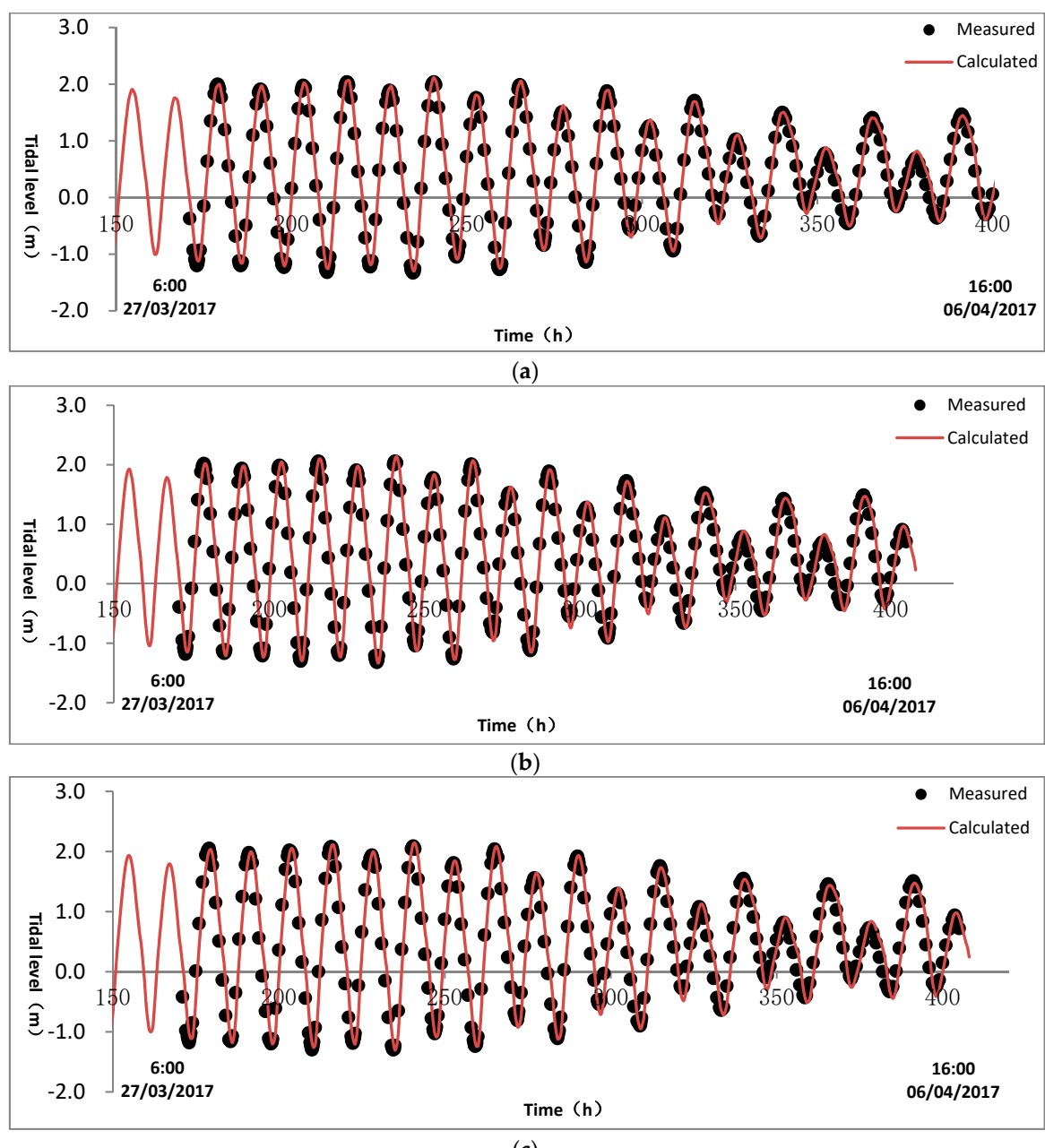

**Figure 5.** Model and measured water level data of observations at (**a**) W1, (**b**) W2 and (**c**) W3.

The flow field calculated by the mathematical model of tidal current in the project area is shown in Figure 6. Taking the high tide as an example, the tide flows into the calculation area from the southeast direction when the tide rises. When through Qushan Island and Changtu Mountain, it was diverted. The water passes through Yanwo Mountain and then enters the engineering sea area. When the tide is ebbing, the water coming from the Gulf of Hangzhou goes around Dayu Mountain to the northwest. Some of the water enters the engineering sea area and further bypasses Yanwo Island and Lipeng Mountain. Through the sea area between Daishan, Changtu and Qushan Islands, it flows into the East China Sea.

*5.2. Validation of Flow Velocity*

Modeled and measured flow velocity data at six stations are presented in Figure 7a–f. Due to space limitations, they are not listed one by one.

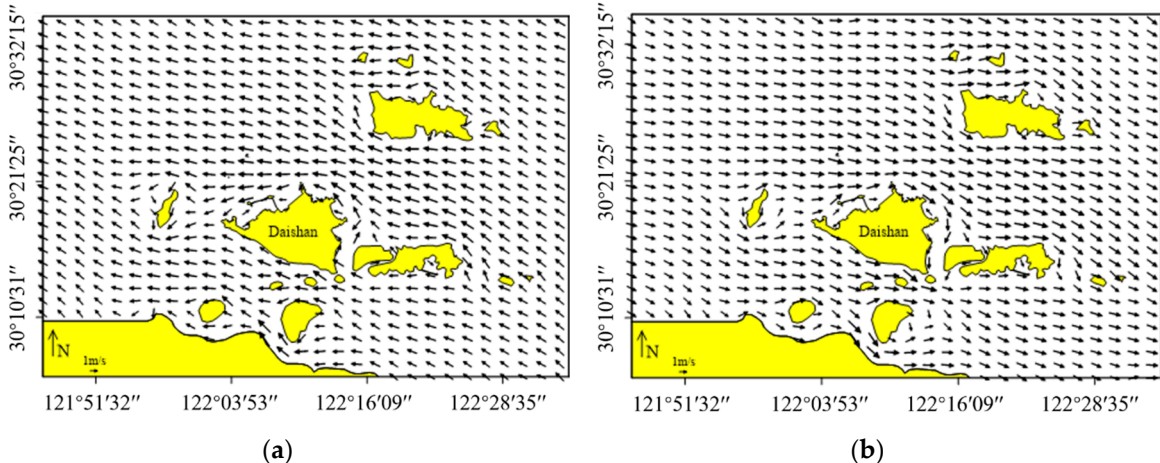

**Figure 6.** Rapid flow chart of tidal fluctuation. (**a**) Flood tide, (**b**) Ebb tide.

**Figure 7.** Modeled and measured flow velocity data at six stations. (**a**) SW1 station, (**b**) SW2 station, (**c**) SW3 station, (**d**) SW4 station, (**e**) SW5 station, (**f**) SW6 station.

## 5.3. Validation of SSC

Regarding suspended sediment concentration, the measured values refer to the hydrological and sediment analysis report of Yanwo Mountain [43]. Figure 8 gives the modeled and measured SSC data from six stations. Verification of the suspended sediment concentration can help us better reflect the transport characteristics of sediment in the engineering area.

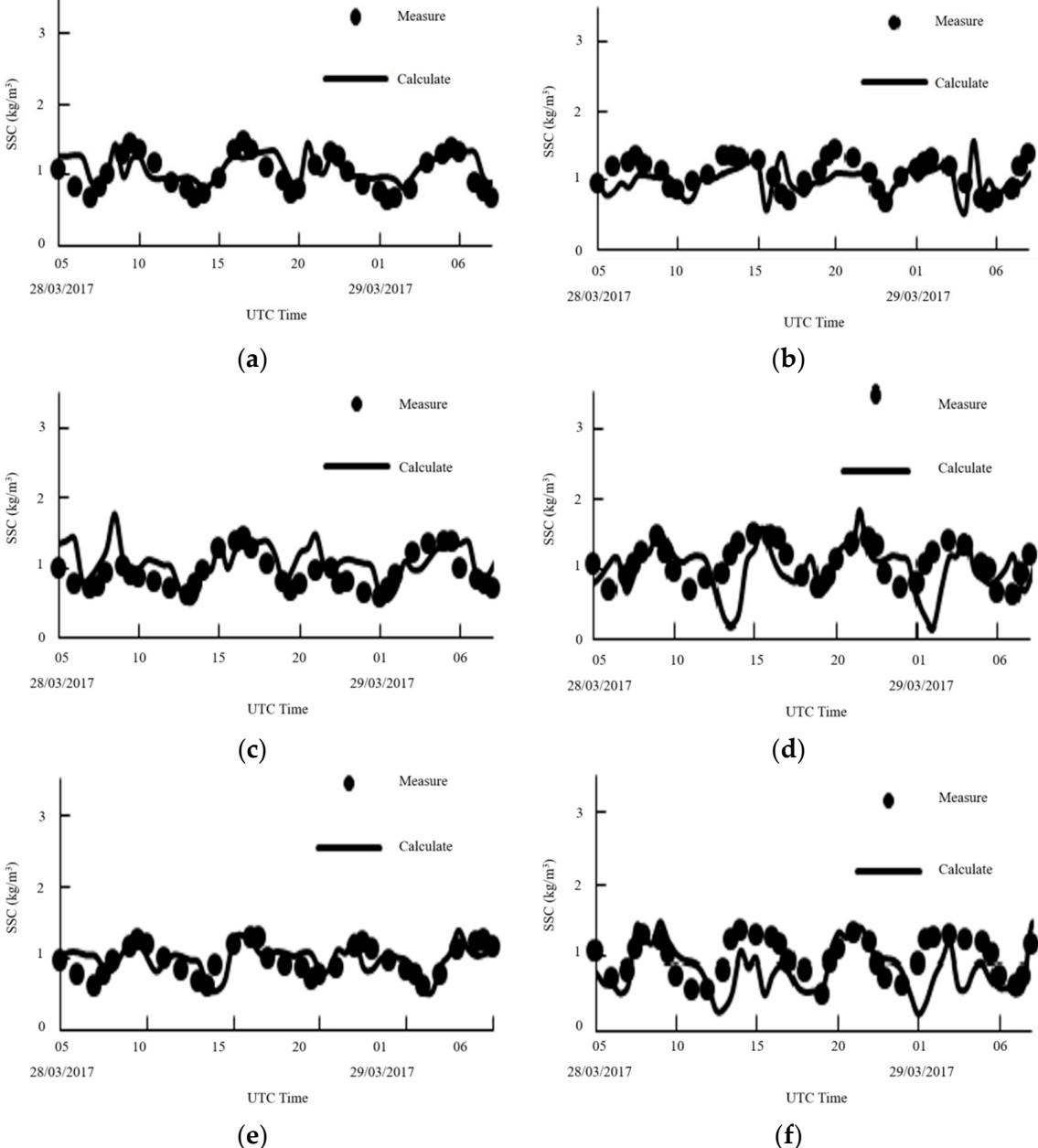

**Figure 8.** Model and measure of SSC data at six stations. (**a**) SW1 station, (**b**) SW2 station, (**c**) SW3 station, (**d**) SW4 station, (**e**) SW5 station, (**f**) SW6 station.

Table 2 shows the correlation coefficients ($R^2$) between the modeled and measured data at stations. The results show reasonable agreement between the model and the data.

**Table 2.** Correlation coefficients ($R^2$) between modeled and measured data at stations.

| Verification Station ID | $R^2$ of Model vs. Measured | | |
|---|---|---|---|
| | Water Level | Flow Speed | SSC |
| SW1 | / | 0.834 | 0.721 |
| SW2 | / | 0.783 | 0.694 |
| SW3 | / | 0.845 | 0.736 |
| SW4 | / | 0.801 | 0.717 |
| SW5 | / | 0.779 | 0.709 |
| SW6 | / | 0.800 | 0.711 |
| W1 | 0.992 | / | / |
| W2 | 0.990 | / | / |
| W3 | 0.990 | / | / |

## 6. Engineering Application Calculation

### 6.1. Project Scenario

According to the general layout of Yanwoshan Island Transportation Terminal Project in the feasibility report of Zhoushan Transportation Planning and Design Institute (2017), the proposed Yanwo Mountain transport terminal is located in the northern part of Daishan Island, Zhoushan, with a geographic location of about 30°20′48″ N and 121°10′23″ E. There are two basic layout schemes, as follows:

Plan I: The breakwater is a fold line type with a length of 1200 m. The length of the break line on the west side of the breakwater is 100 m. Its azimuth angle is N46°~N226°. The length of the east side fold line is 1100 m with azimuth angle N76°~N256°. The east side of the dock and the proposed breakwater are arranged in parallel at an azimuth angle of N76° to N256°.

The land area of the dock is located on the south side of the breakwater and on the southeast side of the proposed wharf. The dock and the land are connected by a through-deck bridge, shown as in Figure 9.

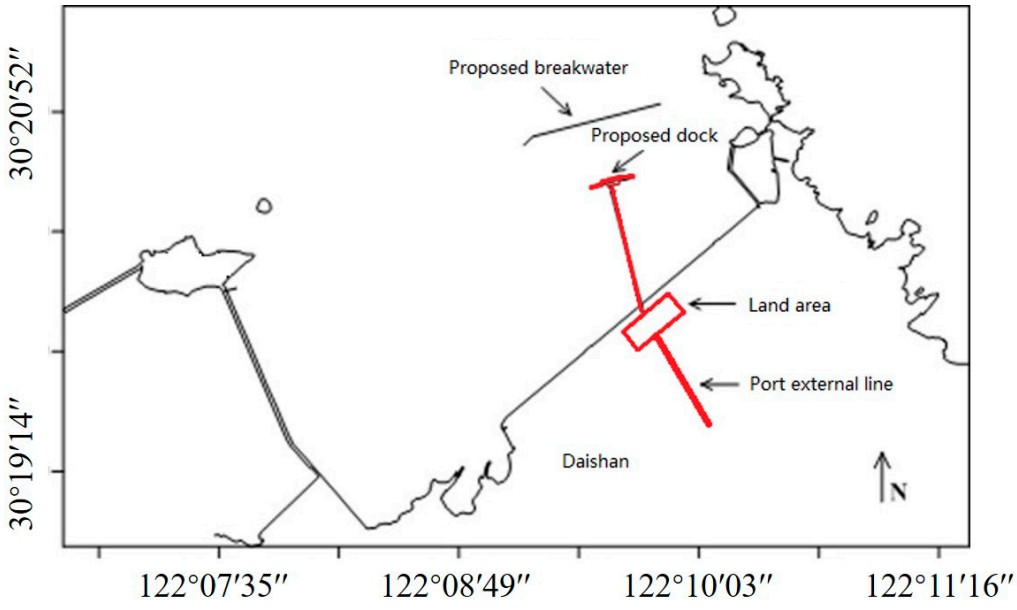

**Figure 9.** The layout of plan I.

Plan II: The axis of the breakwater is arranged in the same way as the first one. The terminal land area is located on the west side of the bird's nest. The terminal and the land area are connected by a through-draft bridge.

The land area is located on the east side of the proposed dock, as shown in Figure 10.

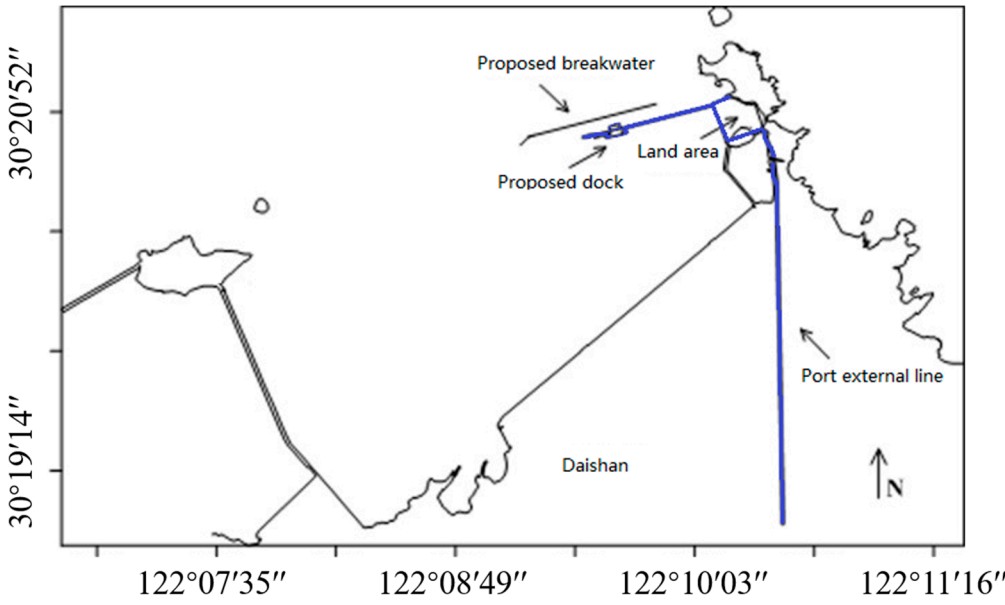

**Figure 10.** The layout of plan II.

The differences between plan I and plan II are shown in Table 3.

**Table 3.** The differences between plans I and II.

| Plan | Plan I | Plan II |
| --- | --- | --- |
| Shoreline | 432 m | |
| Breakwater | 1100 m | 1500 m |
| Dock project | 2 car berths with 3000 gross tonnage (GT) 2 passenger berths with 1000 gross tonnage (GT) (considering 10,000-ton cruise berths) | |
| Rear land area | 100,000 m$^2$ | |
| External wiring road | 1.6 km | 1.5 km |

*6.2. Model Setting of Engineering Area*

According to the layout characteristics and the topographic gradient trend of the proposed breakwater and dock project in the engineering sea area, a small range of models is developed. The rule for determining the calculation range of the small-scale model is that it has no effect on the flow velocities near the boundary of the small-scale calculation area after the implementation of the project. The calculation area contains a sea area of approximately 3583.49 km$^2$.

The large model is the sea area around Daishan Island. The small range of the model is the partial sea area of wharf engineering (Figure 11). To ensure the local flow field calculation of the wharf project conforms to the overall physical characteristics of the tidal current field in Daishan Island, the large-scale model provides the boundary conditions needed by the small-scale model.

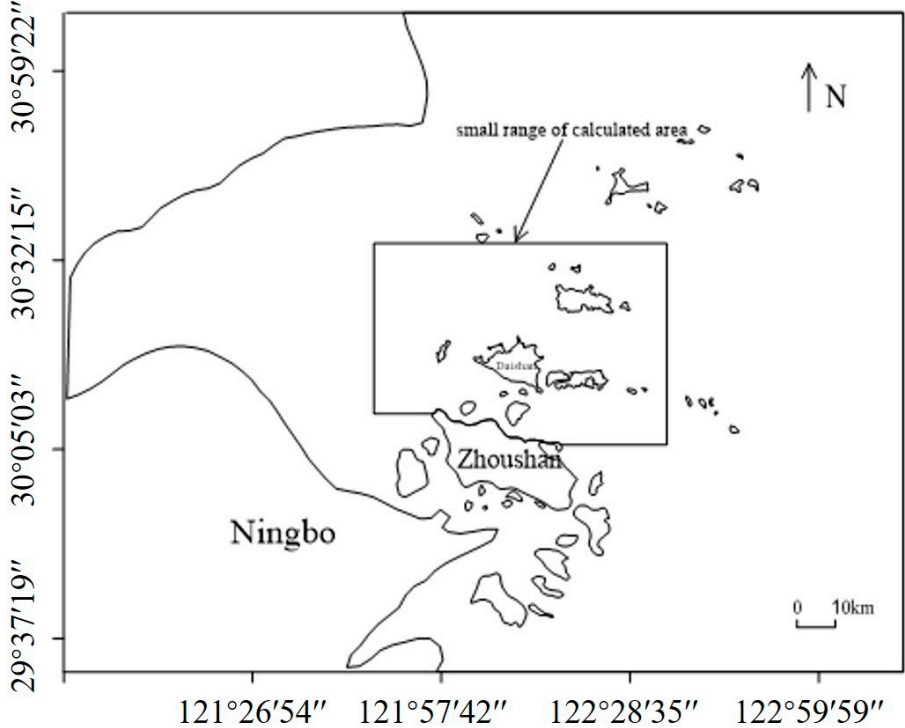

**Figure 11.** The large and small range of calculated area with mathematical model.

The minimum mesh size of the small model is 5 m, the number of grid cells is 34,808 and the grid node is 17,289. The computational domain and mesh generation of this model are shown in Figure 12.

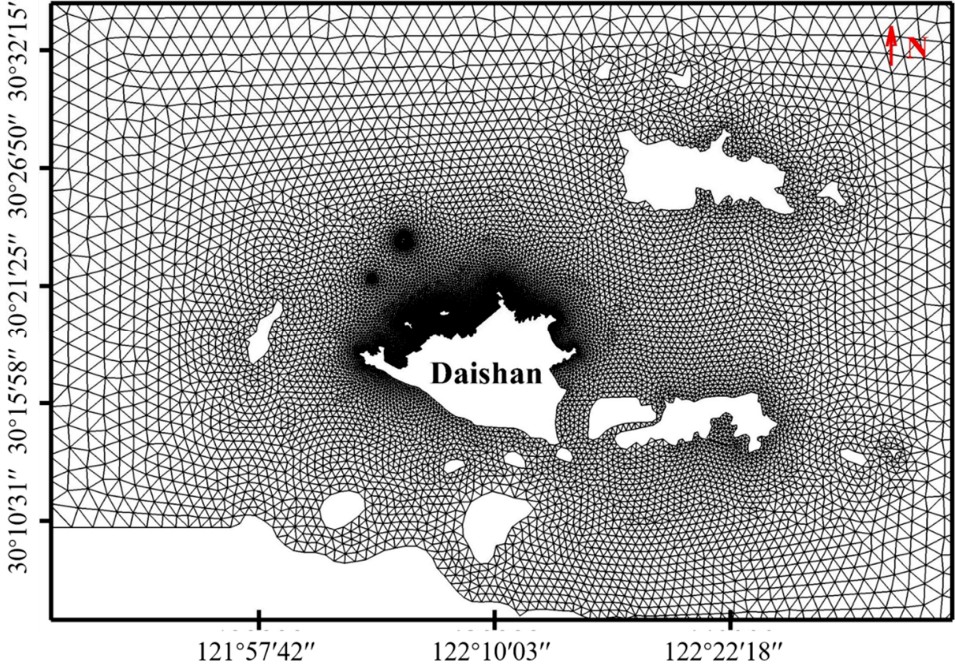

**Figure 12.** A mesh map of the small model area.

## 7. Results and Discussion

### 7.1. The Character of Tidal Movement Changes after Project Implementation

Figure 13 is a diagram showing the maximum flow vectory of flood and ebb tide map after plan I implementation. Figure 14 is a diagram showing the maximum flow vectory map of flood and ebb tide after plan II implementation. It can be seen from the figure that when the tide rises, the water flows from the east side of the Yanwo Mountain to the area of the project. After flowing through Yanwo Mountain, the water flow separates and forms a recirculation zone on the west side of Yanwo Mountain. Due to the blockage of the water flow by the breakwater, the water from the east side gate of the breakwater flow enters the harbor basin and the nearshore waters.

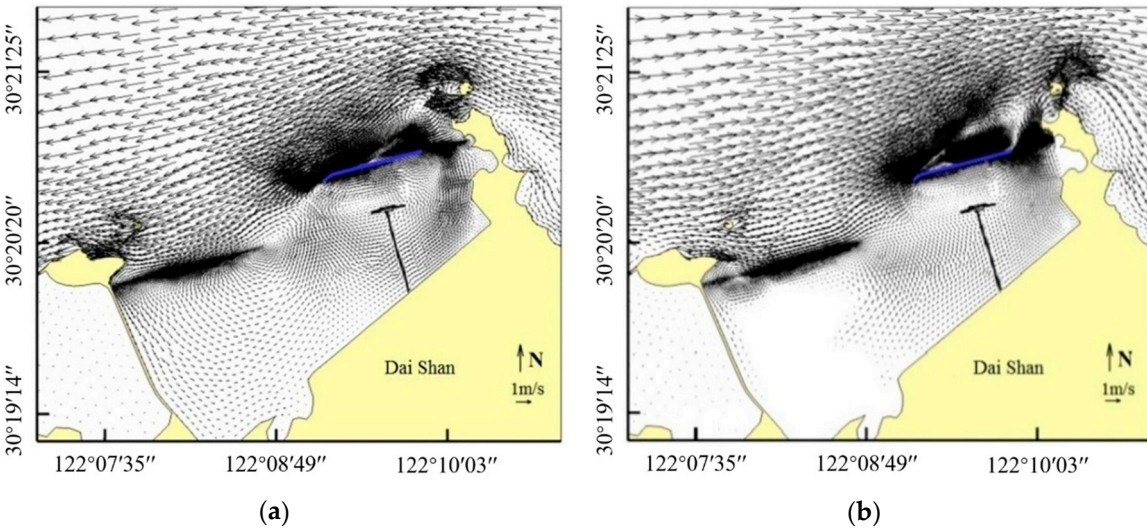

(**a**)  (**b**)

**Figure 13.** The maximum flow vectory of flood and ebb tide map after plan I implementation. (**a**) Flood tide, (**b**) Ebb tide.

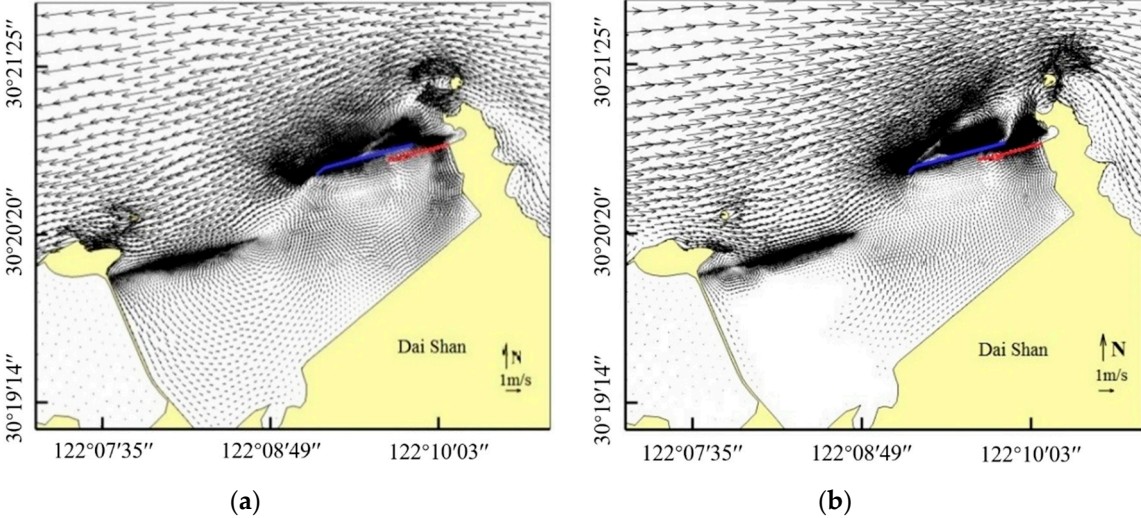

(**a**)  (**b**)

**Figure 14.** The maximum flow vectory map of flood and ebb tide after plan II implementation. (**a**) Flood tide, (**b**) Ebb tide.

The flow rate of the water flow is greatly reduced compared with before the breakwater construction. At the same time, a recirculation zone is formed in the waters of the harbor basin behind the breakwater and the flow rate is less than 0.5 m/s. When the water flows to Dongken Mountain, separation occurs again. The main part of the water continues to move westward and a small part of the water flows to the south side of the coastal waters to form a large area of reflow. When the tide ebbs, the water flows from the northwest side of the project sea area into the sea near the breakwater. When flowing through the Dongken Mountain, the mainstream continues to move to Yanwo Mountain. A small part of the water flows into the shallow waters near the shore and the waters of the harbor basin. Due to the shallow depth of the land boundary, some of the boundary areas are exposed to the water surface during the emergency. The construction of the breakwater results in the separation of water flow when it passes through the breakwater and it forms a recirculation zone in the immediate vicinity of the north side boundary of the breakwater. Due to the complex terrain of the engineering sea area, the flow velocity difference between the north and south sides of the breakwater is large. The flow rate in the north side of the breakwater is relatively large; the rapid flow velocity is 1.75 m/s north of the west bank (about 2 km) and the exit velocity is 2.01 m/s. The flow rate in the shallow waters south of the breakwater is basically less than 0.5 m/s. The trend of the mid-tidal and small-tidal fluctuations is similar to that of the tide but the tidal current is weak and the trend of the tidal current is slightly larger than the tidal flow velocity. Due to the complex terrain near the north side of Daishan Mountain, the deep water gradient and the numerous island reefs, many current vortices will be formed regardless of the rising or falling tide. The tidal current near the engineering sea area is still dominated by the reciprocating flow.

*7.2. Flow Velocity Changes after Project Implementation*

The tidal current near the project sea area is mainly reciprocating flow. The north sea area of Yanwo Mountain and Dongpu Mountain has a large flow velocity exceeding 2.0 m/s. The water depth variation gradient in the coastal waters is large and the flow velocity within the 5 m isobath is small (around 0.6 m/s).

When the tide surged before the project, the average flow velocity and maximum flow velocity of the south and north side of the breakwater showed an increasing trend from east to west. When the tide is ebbing, the average flow velocity and maximum flow velocity on the south and north sides of the breakwater show a trend of decreasing from west to east. The velocity contour map of flood and ebb after plan I implementation is shown in Figure 15. The velocity contour map of flood and ebb after plan II implementation is shown in Figure 16. According to the numerical simulation results of the small-scale tidal current mathematical model before and after the implementation of the first and second schemes, taking the tide as an example, except for the increase of the flow velocity of some stations, the other flow rates are reduced, the maximum flow velocity trend and the average trend of flow rate changes are basically the same.

After the implementation of the breakwater project of each scheme, the impact on the tidal level of the sea near the breakwater is small. After the implementation of the first scheme, the characteristic station with the largest change of the flow rate is increased by 32.88% compared with before the project and the maximum flow velocity of each characteristic station is within 1.0 m/s. The maximum cross-flow velocity of the station at the forefront of the dock is 0.15 m/s, the backflow intensity of the harbor basin is 0.37 m/s and the maximum cross-flow velocity of the characteristic station of the channel is 1.33 m/s. After the implementation of the second scheme, the maximum cross-flow of the characteristic station at the forefront of the dock is 0.12 m/s, the backwater intensity of the harbor basin is 0.49 m/s.

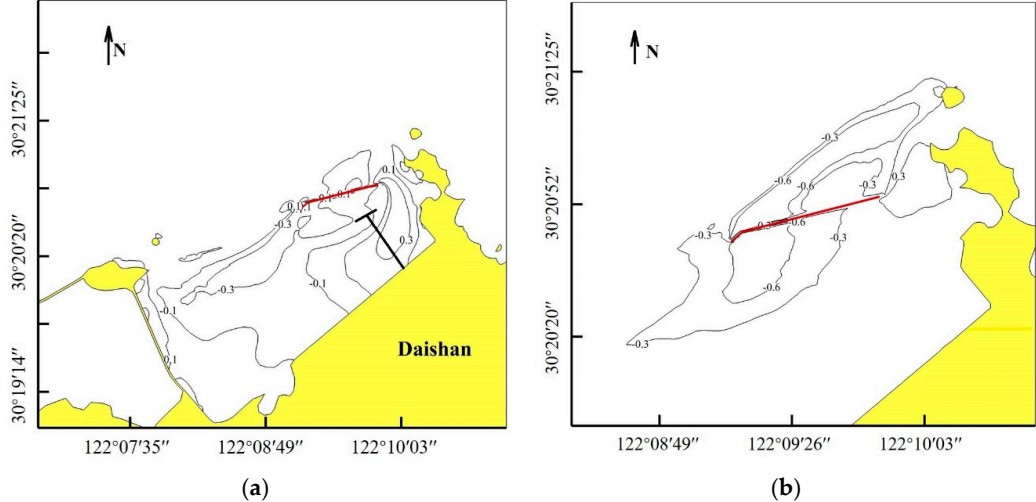

**Figure 15.** The velocity contour map of flood and ebb tide after plan I implementation (unit: m/s). (**a**) Flood tide, (**b**) Ebb tide.

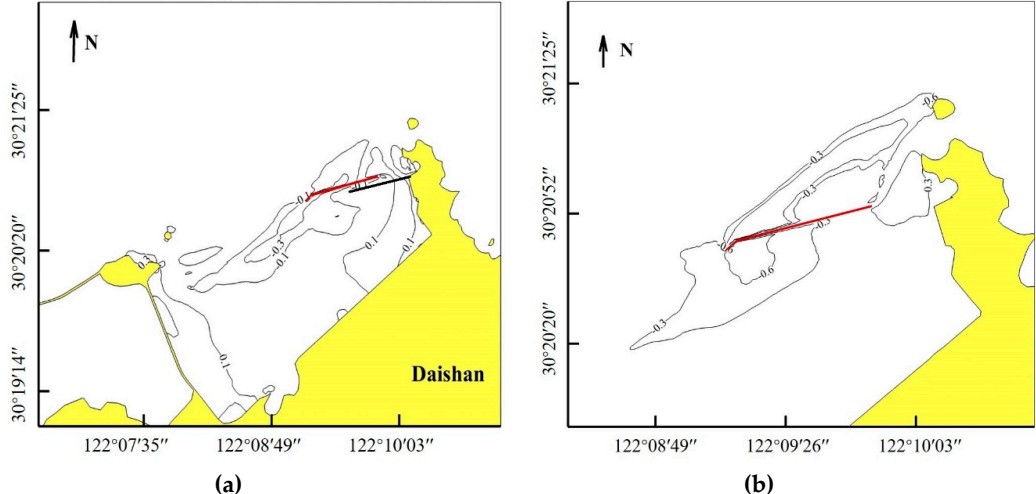

**Figure 16.** The velocity contour map of flood and ebb tide after plan II implementation(unit: m/s). (**a**) Flood tide, (**b**) Ebb tide.

*7.3. The Change of Sediment Deposition after Implementation*

The siltation strength of each major scheme is shown in Figure 17, where "positive" means deposition and "negative" means erosion. The annual maximum sedimentation intensity and annual siltation amount in the channel terminal area after implementation of different schemes are shown in Table 4. It can be seen from the table that after the implementation of plan I, the sediment erosion intensity in the sea near the breakwater has changed. Because of the angle between the longitudinal axis and the isobath of the breakwater, the flow velocity decreases when the water flows near the breakwater and the SSC falls and deposits. At low tide, as a result of the construction of the breakwater, the flow velocity in the backflow area decreases and the sediment deposition strength increases. The bottom bed on both sides of the recirculation zone has a slight scouring phenomenon. Due to the water binding action of the breakwater on the east entrance of the breakwater, the velocity of the head of the breakwater increases and the strength of sediment erosion increases. After the implementation of the breakwater project, dredging engineering will cause the navigation channel and the harbor basin to form silt. The maximum siltation intensity is located at the curved section of the channel. The arrangement of the plan II breakwater is the same as that of plan I. Only the land area is set near

the east side of the bird's nest mountain, which has little effect on the sediment erosion intensity of the project area. The change trend of the erosion and siltation intensity is similar to that of plan I.

**Table 4.** The maximum siltation intensity and the total annual sedimentation volume of waterways and harbor basins after each plan implementation.

|  | Waterway and Harbor Water Area ($10^4$ m$^2$) | Maximum Deposition Strength (m/year) | Annual Siltation ($10^4$ m$^3$/year) |
|---|---|---|---|
| Plan I | 26.12 | 0.71 | 8.65 |
| Plan II | 29.41 | 0.69 | 9.53 |

It can be seen from the above analysis that the total annual siltation of the navigation channel and the harbor waters after the implementation of the first plan is 86,500 m$^3$/year In the comparison of plan I, the sedimentation volume of the navigation channel and the harbor basin caused by the arrangement of the plan II breakwater is 95,300 m$^3$/year.

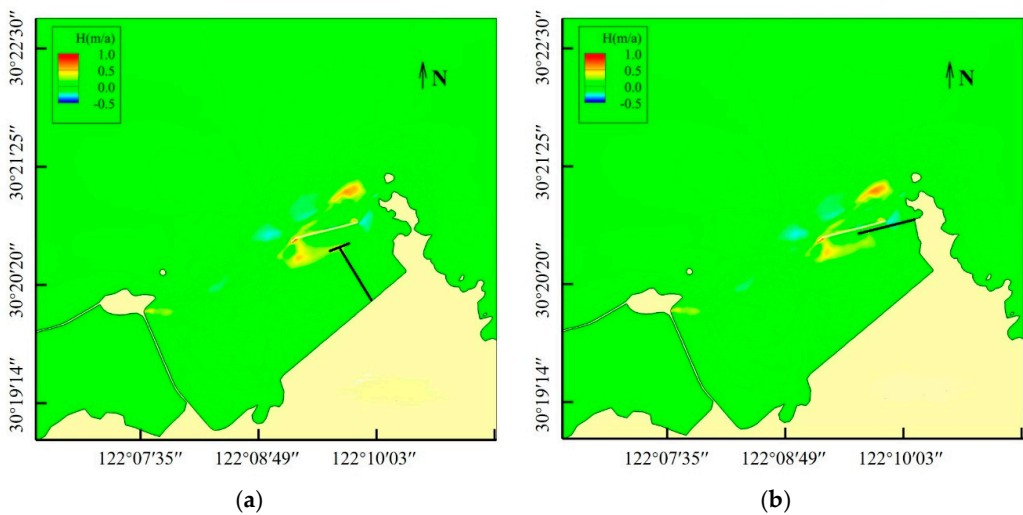

(**a**)  (**b**)

**Figure 17.** Distribution of siltation intensity before and after different plans' implementation. (**a**) Plan I, (**b**) Plan II.

*7.4. Discussion of Crosscurrent Velocity*

The tidal current station at the entrance of the harbor with plan I and plan II is shown in Figure 18. The tidal current stations are arranged separately—C1–C5—to calculate the cross flow speed. Table 5 shows pre-engineering cross flow speed statistics; Table 6 shows cross flow speed statistics after engineering.

**Table 5.** Pre-engineering cross flow speed statistics.

|  |  |  | C1 | C2 | C3 | C4 | C5 |
|---|---|---|---|---|---|---|---|
| Plan I | peak cross flow speed (m/s) | tides rise | 0.96 | 0.90 | 0.84 | 0.80 | 0.81 |
|  |  | tides fall | 1.47 | 1.43 | 1.26 | 1.09 | 1.04 |
|  | Duration (h) (cross flow speed > 0.5 m/s) | | 17.50 | 17.00 | 15.50 | 13.50 | 13.50 |
| Plan II | peak cross flow speed (m/s) | tides rise | 1.02 | 0.97 | 0.90 | 0.81 | 0.78 |
|  |  | tides fall | 1.32 | 1.49 | 1.43 | 1.24 | 1.08 |
|  | Duration (h) (cross flow speed > 0.5 m/s) | | 17.50 | 17.50 | 17.00 | 16.00 | 13.50 |

**Table 6.** Cross flow speed statistics after engineering.

| | | | C1 | C2 | C3 | C4 | C5 |
|---|---|---|---|---|---|---|---|
| Plan I | peak cross flow speed (m/s) | tides rise | 1.07 | 1.11 | 1.01 | 0.56 | 0.24 |
| | | tides fall | 1.33 | 1.20 | 0.96 | 0.74 | 0.56 |
| | Duration (h) (cross flow speed > 0.5 m/s) | | 17.50 | 15.50 | 12.50 | 6.00 | 2.50 |
| Plan II | peak cross flow speed (m/s) | tides rise | 0.99 | 1.07 | 1.11 | 0.9 | 0.34 |
| | | tides fall | 1.40 | 1.34 | 1.17 | 0.89 | 0.65 |
| | Duration (h) (cross flow speed > 0.5 m/s) | | 17.50 | 17.00 | 15.50 | 11.50 | 4.00 |

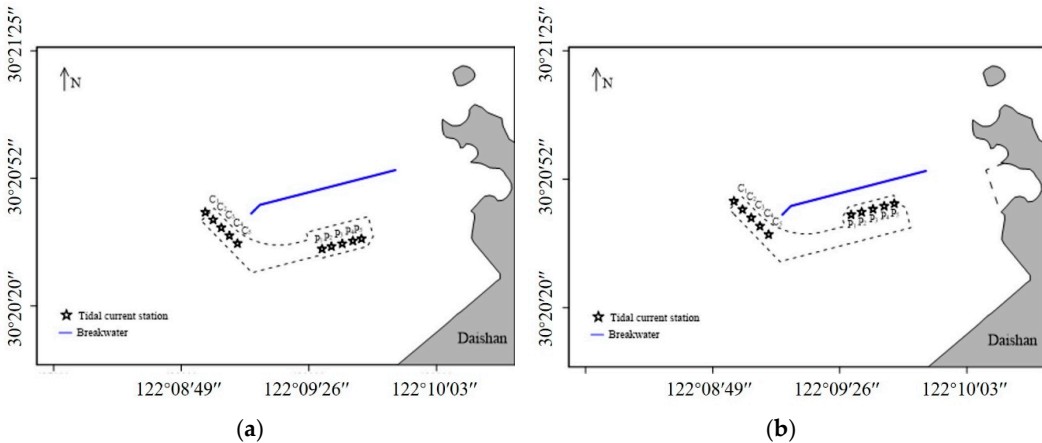

(**a**)  (**b**)

**Figure 18.** The tidal current station of the entrance of harbor with plan I (**a**) and plan II (**b**).

It can be seen that the layout of the newly planned harbor has a major impact on the cross flow speeds in the entrance channel. The peak cross flow speed of plan I increases from 0.96 m/s to 1.11 m/s. The peak cross flow speed of plan II increases from 1.02 m/s to 1.11 m/s. The peak cross flow speed occurs near the entrance of the new planned harbor. This will affect the navigation safety of ship entering or leaving the harbor.

*7.5. Discussion of the Consequences of Choosing Plan I or Plan II*

The advantages and disadvantages of choosing plan I or plan II are shown as Table 7.

**Table 7.** The advantages and disadvantages of choosing plan I or plan II.

| | Plan I | Plan II |
|---|---|---|
| Advantages | 1. Due to the block of Yanwoshan, the length of the breakwater will be reduced. So, the cost of the dock project will be reduced 2. The east side of the land area is the wave-breaking area and the dock is stable | 1. The water depth at the dock is better than plan I |
| Disadvantages | 1. The overall water depth of the terminal is worse than that of the second option and maintenance dredging is required every year | 1. The port is covered by a breakwater and the breakwater will be longer than plan I 2. The front edge of the dock is affected by the tidal current, which is not conducive to the stability of the dock |
| Investment estimate | Almost 599.10 million yuan | Almost 671 million yuan |

## 8. Conclusions

According to the planning of the Daishan dock project, combined with the small-scale mathematical model that we established, results and analysis have led to the following conclusions.

After the implementation of the Daishan dock project, it did not have a great impact on the overall flow pattern in the northern part of Lushan Mountain. The trend is still not much different from the original and the falling tide is faster than the rising speed. However, the flow velocity around the breakwater project changed. At the same time, due to the existence of the breakwater, the backflow occurs around the breakwater and the flow velocity changed significantly.

The implementation of the breakwater and the excavation of the channel pool caused a certain degree of change in the erosion and siltation conditions in the sea around the project area. At the east gate of the breakwater, due to the water-splitting effect of the breakwater, the flow rate and sediment carrying capacity increased. Thus, an erosion state has existed. The flow velocity decreases and the siltation state appeared in the south and north sides of the breakwater and the channel harbor area.

In summary, the Daishan dock project has a certain impact on the water and sediment environment in the Daishan sea area but the impact is mainly in the sea area near the breakwater; the tidal movement and sediment transport laws of the whole sea area has not changed.

Numerical simulation has been presented in this article for predicting strong sediment transport in the sea near the breakwater project. The hydrodynamic component of the model employs a non-structured, triangular grid system. In the application on Daishan, the performance of the hydrodynamic model has been satisfactorily validated against observed variations of water level, flow velocities and SSC at available measurement stations. Coupled with the hydrodynamic model, a sediment transport model has been developed and tested.

The area of the waterway and harbor basin of plan II is larger than in plan I, so there will be a case where the total sedimentation amount of plan II is greater than that of plan I in the second year. However, the water area and harbor basin of plan II increased by 12.59% compared with plan I, while the annual total siltation of plan II increased by 10.17% compared with plan I. At the same time, the maximum annual sedimentation intensity of plan II is less than that of plan I. The comprehensive evaluation is that plan II is better than plan I. However, the design width of the channel should be increased and the angle between the axis of the channel and the flow direction of the channel near the channel gate should be reduced to ensure the safety of the ship.

In future studies, a sensitivity analysis should be performed in order to discuss the effect of different parameters on the simulations. In addition, in future work, global sensitivity analysis or robust sensitivity analysis should be applied to identify critical model input parameters and quantify the impact of uncertainty on model outputs, due to global sensitivity analysis can provide reliable diagnostic insights that are robust to highly uncertain future conditions [44–46]. Besides, wave motion will be considered in next paper [47–49].

**Author Contributions:** Y.L. has supervised the research, finished the first draft of the manuscript; Z.S., G.P. and R.L. edited and reviewed the manuscript and contributed to the model construction and verification; X.F., P.C. and H.H. edited and reviewed the manuscript.

**Funding:** This work was supported by the National Key R & D Program of China (Grant No. 2018YFB0505500, 2018YFB0505502).

**Conflicts of Interest:** The authors declare no conflict of interest.

### Notation

The following symbols are used in this paper:
$d$ = the total water depth
$\eta$ = water level
$h$ = hydrostatic depth
$u$ = current velocity at the $x$ direction
$v$ = current velocities at the $y$ direction

$g$ = acceleration of gravity
$f$ = Coriolis parameter
$C_z$ = chezy coefficient
$\varepsilon_{xx}, \varepsilon_{xy}, \varepsilon_{yx}, \varepsilon_{yy}$ = the coefficient of eddy viscosity at $xx$, $xy$, $yx$, $yy$ direction
$F_c$ = suspended sediment flux
$\varepsilon_x$ and $\varepsilon_y$ = diffusion coefficient
$\alpha$ = the settlement probability of sediment
$\omega$ = settling velocity of SPM
$M$ = the coefficient of scouring
$\tau_b$ = bed shear stress
$\rho$ = the water density
$U_*$ = the shear velocity
$\tau_e$ = the critical shear stress for sediment resuspension in $\tau_e = \rho U_{*e}^2$
$U_{*e}$ = the critical shear velocity for sediment resuspension
$\tau_d$ = the critical shear stress for sediment deposition in $\tau_d = \rho U_{*d}^2$
$U_{*d}$ = the critical shear velocity for sediment deposition
$\gamma_0$ = dry density of sediment
$Z_b$ = bed elevation
$\Gamma$ = open boundary of waters
$C^*(x, y, t)$ = suspended sediment concentration
$x$, $y$ = physical coordinates
$t$ = time

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
