# Peer review of "Modeling Hydro-Dynamics in a Harbor Area in the Daishan Island, China"

_water, doi:10.3390/w11020192_

Round 1
Reviewer 1 Report
General comments to the authors
In the present paper a numerical model has been presented to analyze the impact on hydrodynamics and erosion/deposition process of two different engineering projects. The model has been adopted to choose between the two engineering proposals.The paper deals with an interesting topic and it shows how a numerical model should be developed to be applied to a real engineering problem.
However, the manuscript suffers from some shortcomings that need to be addressed before it can be accepted. Therefore, I recommend a major revision before publishing the paper, also considering the detailed comments below.
Detailed comments
English is not always fluid and correct. So, I suggest the paper be double checked by a professional English writer.
The authors completely neglect the wind action and hence the wave motion. This is one of the main concerns related to the present paper. This must be justified also with respect to recent references, dealing with this topic:
Gad, F.-K.; Hatiris, G.-A.; Loukaidi, V.; Dimitriadou, S.; Drakopoulou, P.; Sioulas, A.; Kapsimalis, V. Long-Term Shoreline Displacements and Coastal Morphodynamic Pattern of North Rhodes Island, Greece. Water 2018, 10, 849.
Petti, M.; Bosa, S.; Pascolo, S. Lagoon Sediment Dynamics: A Coupled Model to Study a Medium-Term Silting of Tidal Channels. Water 2018, 10, 569.
Sadio, M.; Anthony, E.J.; Diaw, A.T.; Dussouillez, P.; Fleury, J.T.; Kane, A.; Almar, R.; Kestenare, E. Shoreline Changes on the Wave-Influenced Senegal River Delta, West Africa: The Roles of Natural Processes and Human Interventions. Water 2017, 9, 357.
Matteo Postacchini, Aniello Russo, Sandro Carniel, and Maurizio Brocchini (2016) Assessing the Hydro-Morphodynamic Response of a Beach Protected by Detached, Impermeable, Submerged Breakwaters: A Numerical Approach. Journal of Coastal Research: Volume 32, Issue 3: pp. 590 – 602.
Furthermore, it is generally necessary to extend and update the references, which should include the developments in recent years.
Line 79. SPM has not been defined yet.
Lines 90-95. All names of locations and geographical entities must be shown in a figure. In figure 1 only Daishan Island (or Dai Shan Island??) is shown. Zhejiang province, Yangtze River estuary, Hangzhou Bay, Donghai, Zhoushan Island and Zhoushan (province?? region??) are missing. A larger map of the east coast of China is needed, more or less like figure 3 from Shanghai to Wenzhou, to contextualize the study area. And then a more detailed map of the focus of the study area. Please, remember that all locations mentioned in the whole paper must be shown in a map.
Lines 101-104. Repeated (see lines 94-97).
Line 106. “The maximum annual precipitation”: are you sure? The average annual precipitation is declared to be 873.6 mm (line 105); it is not possible that the maximum annual one is 128.6 mm.
Lines 113-116. All directions are expressed in degrees north, isn’t it? This should be mentioned in the paper.
Line 122. The maximum/minimum “vertical sediment concentration” is the vertical averaged value, isn’t it? This should be mentioned in the paper.
Is Figure 2 mentioned in the text? This must be checked. Moreover, figure resolution must be improved.
Lines 140-171. All symbols in the text should be in-line with the text and not “above” the line. This must be fixed.
Lines 144-145. Symbols for “rotational-angular velocity of the Earth” and “local latitude” do not appear in the equations. Thus, they must not be presented in the text.
Line 152. Symbol omega is not explained in the text.
Lines 153-158. Parameters as alfa, M, U*e and U*d are very important in the model and their value can strongly influence the results. The values assumed by the authors in the present paper must be declared and discussed, also with respect to recent references, dealing with this topic:
Bosa S., Petti M., Pascolo S. (2018) Numerical Modelling of Cohesive Bank Migration, Water 10, 961.
Hu X., Yang F., Song L., Wang H. (2018) An Unstructured-Grid Based Morphodynamic Model for Sandbar Simulation in the Modaomen Estuary, China, Water 10(5), 611.
Xie Q., Yang J., Lundström S., Dai W. (2018) Understanding Morphodynamic Changes of a Tidal River Confluence through Field Measurements and Numerical Modeling. Water 10, 1424.
Line 182. Hydrodynamic data as model input: is this referred to model/grid boundary? This must be better specified.
Line 198. Please, change “calibrate” with “validate”: as far as I understand, the model has not been calibrated, but is has been validated only.
Figure 5. Figure resolution must be improved. Moreover, the “engineering stations” should be better presented (what do they measure?); in figure 2 they are almost not visible.
Lines 214-219. Again, all names of locations and geographical entities must be shown in a figure.
Reviewer 2 Report
REVIEW:
Numerical simulation to analyze sediment transport near the dock and breakwater project in the sea near Yanwoshan, Daishan Island
This manuscript concerns an extension of a sediment transport numerical model were the possibility of modelling suspended transport has been included. Results are compared with field data. Based on such comparison the design of the dock and breakwater structure was proposed. The research herein presented is certainly within the scope of the Journal of Water. However, the manuscript organization, the incompleteness and the lack of clarity/novelty in some parts of the text should be amended prior to publication. My current decision is major revision and I reckon the author to take into consideration all the queries listed below before the manuscript could be considered for publication.
List of comments:
- In the title the authors should specify that this is a case study. This is not pure novel since the numerical model was already developed and an extension was included.
- In line 43 the authors when listing the sediment transport numerical models the authors should also acknowledge the following references:
Assessment of the performance of numerical modeling in reproducing a replenishment of sediments in a water-worked channel. C. Juez, E. Battisacco, A. J. Schleiss, M. J. Franca. ADVANCES IN WATER RESOURCES. 2016. DOI: 10.1016/j.advwatres.2016.03.010.
One-Dimensional Riemann Solver Involving Variable Horizontal Density to Compute Unsteady Sediment Transport. C. Juez, J. Murillo, P. García-Navarro. JOURNAL OF HYDRAULIC ENGINEERING. 2015. DOI: 10.1061/(ASCE)HY.1943-7900.0001082.
In the abstract it is said that 3D velocity measurements were performed. However, the results did not show such measurements. Could the authors really show the velocities in the three directions?.
- Lines 48-66 I would shorten that part dramatically. These lines seem to me like the logbook of a research group, where the successive goals achieved by the group are outlined. I don`t think they bring any useful information for the reader.
- In line 144 the rotational-angular velocity of the Earth is mentioned however is not included in the set of equations previously mentioned. Please, check that out.
- In equations 2 and 3 the wind force is not included and for tidal wave simulations is relevant, since it dominates the formation of ripples in the bed sea. Could the authors elaborate on this matter?.
- In equation 4 the extra density added by the sediments in suspension is not included in the equation since it is assumed that density is constant (i.e. the water density) and it can be simplified. Could the authors elaborate on this matter?.
- Figures 5 and 7. These figures displayed the water level and flow velocities of the tidal movement. Nothing is said about the calibration of the eddy viscosity and the mesh size. From my experience, the accurate prediction of water levels and velocities in oscillatory flows is highly impacted by the eddy viscosity and mesh size, which have to be precisely tuned and controlled. Could the author elaborate on this matter?.
- The quality of the text in Figure 7 should be improved. Currently, it is hard to read.
- Make a zoom in Figure 16, otherwise it is too small.
Reviewer 3 Report
Dear Authors,
your research is interesting and, showing the potential of a hydro-morphodynamic model in forecasting the effects of engineering structures, suggest that scientists and practitioners should work more closely in the future.
However, despite the promising framework, the presentation of the work is quite poor, and affect the impact that such kind of research can have. As you find in my following comments, I suggested some points to address in revising the paper, but I would like to stress the need of working on the paper structure and the English language.
Section 3 presents the main equations used in the model, but do not really detail the development of such a model. What is not clear to me is: is the model already available and previously adopted, or did you develop a new one from the scratch? I would like to see some discussion on this.
In Section 4 you describe the model setup. In choosing this setup, did you perform some sensitivity analysis? Can be interesting to present also the procedure used to define the best parameters for the simulations.
Section 6 requires a better description of the model setup, discussing in a proper way the differences between the two plan and the reasons for going for the plan I or II (subsection 6.1), as well as the adopted boundary conditions and dimensions of grid cells (subsection 6.2). As said before, for deciding the spatial resolution did you perform a sensitivity analysis, trying different options?
Looking at the title of Section 7, one expects to see some discussion of the results aside from their presentation. Reading the section, however, the results are just described, without any discussion about their reliability, the possible improvements or the consequences of choosing plan I or plan II. I am looking forward to seeing a proper discussion of your results.
Please, involve a colleague proficient in English, because there are several sentences which are not clear enough. Indeed, involving a native-English speaker could be very helpful in improving the overall quality of the paper and in pointing out the strengthens of the research, actually hindered behind a not clear writing style.
In the following, you can find some detailed comments that should be addressed before the resubmission.
Title
Avoid repetitions and go for a shorter title, like “Modelling hydro-morphodynamics in a harbour area in the Daishan Island, China”
Abstract
line 21-22: what do you mean with “reasonable solution”? I suggest deleting this sentence.
l. 22-25: rewrite the sentence because it is difficult to understand, especially in its last part
Keywords
l. 27: what do you mean with “Numerical accumulation”
1. Introduction
l. 37-39: what do you mean? I cannot catch the subject of this sentence
l. 47: delete the sentence “Lots of research results had been regard” or rewrite the paragraph to better incorporate it
l. 70-73: rewrite the sentence
l. 77: use “a numerical model” instead of “the numerical simulation”
l. 78: what dock project?
l. 79: what is SPM? You should define it
l. 83-88: there are several English errors, especially as for the structure of the sentences. Please, rewrite this part
2. Study area and data used
l. 90: delete “under investigation”. If you are speaking about a study area, is obvious that it is under investigation…
l. 91: delete “which”
Figure 1: add more details to the figure, like a graphic scale, coordinates, north arrow, etc. Consider that the Journal as an international audience; therefore, please avoid Chinese symbols.
l. 94-104: avoid repetitions. Please, double-check the manuscript before the submission…
l. 114: please add the units for defining the flow direction
l. 113-116: I suggest adding a table summarizing these quantities, giving that reading these sentences is quite annoying
Figure 2: is this figure necessary? Do you refer to it somewhere in the text? Moreover, the quality is very low, and the figure is illegible
3. Model Developing and Governing Equations
l. 125: I suggest changing the title in “Governing Equations”, because you just present the main equations without really developing a model
l. 126: delete “Subsection”
l. 130-132: this should be a single sentence: change the point after “process” with a comma
l. 150: ε are the diffusion coefficients or the coefficients of eddy viscosity (l. 146)? Adding a notation list can be helpful in avoiding misunderstandings in reading the equations
Please, correct the several English errors that are present in this section. Some sentences are simply impossible to understand.
4. Model setup
l. 181-182: rewrite this sentence
l. 182: what article? Probably moving the citation just after the name of the authors could be useful for the readers
l. 183: “escriptions” should be read as “descriptions”? Again, involve an English-native speaker, or at least use a spell checker
l. 190: from 21 March “to” 7 April
l. 190: add the units of the roughness
5. Validation of the model
l. 197-201: you say that model verification is very important, and then you perform the verification based on only six days. Are you sure that such values are a good representation of a normal behaviour? Moreover, am I wrong, or you performed simulations and verifications over the same period? Usually, two different periods are used for calibration and validation.
l. 203: you can add a table summarizing the statistics of each station. In this manner, the reader can understand the reliability of the model in representing all the observed data
l. 223: is this title wrong? You described the validation of the water level in the previous subsection… Probably the correct title is “Validation of water velocity”
l. 224: use “velocity” instead of “speed”
Figure 7: are you able to read the time of the x-axis? Please, improve the figure
l. 238: please, add a reference when you are speaking about a report
l. 239-240: what is the meaning of this sentence?
6. Engineering application calculation
l. 258: use “with” instead of “which”
Figure 9 and Figure 10: please, consider combining these figures in a single one, using some colours to distinguish between the two plans
l. 284: pay attention to singular/plural. You can say that “the number of the grid cells is...” or “the grid cells are…”
7. Results and Discussion
l. 301: delete “flows”
Figures 13 and 14: the captions should be updated, better defining the differences between the two panels. In the text you said that these figures report the current and the modelled tide fluctuations: please, update the caption accordingly
l. 331-335: rewrite these sentences, because it is not clear what are the results that you want to discuss
Figures 15 and 16: for the captions, see my comments above. In addition, please specify the units of the isolines
l. 342-344: usually, in representing bathymetric changes, positive values mean a deposition and negative values an erosion with respect to the previous situation. If I correctly understood the text, it is the same in your case. Please, simplify the sentence if my interpretation was correct, or change it for clarifying my doubts
Table 1: are you sure that the reported values are correct? There is an order of magnitude of difference between the text (l. 361-363) and the table
l. 368: “Results” of what? Section 7 is already entitled “Results and Discussion”
l.369-388: do you think that repeating the results could be helpful for the readers? Please, delete the outcomes already reported in the text, and integrate the missing ones in the appropriate sections
8. Conclusions
l.392-396: you say that the project “did not have a great impact” but also that “the flow velocity…varies greatly”. Can you please discuss better this fact? I am not sure to correctly catch your conclusion, but, in my opinion, if something changes greatly it has a great impact
l.420-422: from which analysis do you derived such conclusions? I am not able to find it in the text. Can you add some details in Section 7?
Round 2
Reviewer 1 Report
The manuscript is a revision of previous submission no water-411099.
I appreciate the time the authors have spent to address all my comments. Nevertheless, there are some inaccuracies that must be fixed. Therefore, I recommend a minor revision before publishing the paper.
Table 1. The authors should better explain what they mean with “bottom quality” in the “Observation” column.
“Notation”. Some symbols in the text are still “above” the line and not “in-line” with the text. Take for example lines 450 or 452: symbols d and h are not at the same level as the text. On the contrary, lines 453-455 are ok. This must be fixed.
Nevertheless, the main concerns are related to the numerical model:
1. Paragraphs 3.1 and 3.2. As an answer to Reviewer 3, the authors declare that Mike21 has been used to solve the hydrodynamic problem. This was rather a surprise. From the previous version of the paper, I understood that the model was an in-house model. Anyway, how have equations (4) and (6) been solved? Usually, they are solved together with the hydrodynamic equations (1), (2) and (3), in a coupled way. So, I guess that Mike 21 has been applied also for the sediment transport problem and not only for the hydrodynamic problem. Is this correct? This should be specified in the text. Moreover, if this is the case, the authors should not speak of “model development” nor here nor in the Conclusions (line 425), because this is an application only.
2. In the reply to Reviewer 2 and to me the authors write that “the water depth in the study area is large, and the influence of waves on the bottom sand is small”. I agree with them, but the area they are considering is very close to the coast (see Figure 9 for instance). Is really the water depth large enough to neglect the influence of wind waves? Do you know the wave parameters typical of this area, to say that they are not relevant? Can you please add a figure with a contour of water depth (like Figure 3) on the area depicted in Figure 9?
Author Response
please see attachement

Reviewer 2 Report
The authors considered all my queries.
Reviewer 3 Report
Dear Authors,
thank you for providing me with a point by point response to my comments.
Even if several comments were addressed, your research is still lacking in some points.
The most important one regards the adopted model:
- you used some literature parameters for your case study. Are you sure that such coefficients give the corrected results? You should firstly perform a sensitivity analysis, and then compare future scenarios, and not vice versa as you are intended to do.
- I was not able to catch your answer: are the calibration and the validation made over the same period? Probably it is a matter of language, but, please, give me a clear response.
In one of your answers, you said "It should be a Manning's n. So, it has no unit". Are you sure? Please, double-check the units of the roughness, as described by Manning.
